# Evaluation of the First-Year Data from an HPV Vaccination Van Program in South Carolina, U.S.

**DOI:** 10.3390/jcm12041362

**Published:** 2023-02-08

**Authors:** Marvella E. Ford, Kathleen B. Cartmell, Angela M. Malek, Phuong Nhi Thi Le, Chloe Keeve, I’Ayana Sanders, Jerlinda Ross, Melanie Slan, Joan McLauren, Mina Platt, Ellen Gomez, Jessica Zserai, Beth Poore, Christina Cody, Victoria Ladd, Mary Spanos Beattie, J. David Sudduth, Kapri Kreps, James R. Roberts

**Affiliations:** 1Department of Public Health Sciences, College of Medicine, Medical University of South Carolina, 135 Cannon Place, Charleston, SC 29425, USA; 2Department of Public Health Sciences, Clemson University, 519 Edwards Hall, Clemson, SC 29634, USA; 3College of Medicine, Medical University of South Carolina, 171 Ashley Avenue, Charleston, SC 29425, USA; 4Department of Biological and Physical Sciences, South Carolina State University, 300 College Street, Orangeburg, SC 29117, USA; 5Department of Obstetrics and Gynecology, College of Medicine, Medical University of South Carolina, 10 McClennan Banks Dr., Charleston, SC 29425, USA; 6Hollings Cancer Center, College of Medicine, Medical University of South Carolina, 86 Jonathan Lucas Street, Charleston, SC 29425, USA; 7Immunization Division, South Carolina Department of Health and Environmental Control, 2600 Bull St., Columbia, SC 29201, USA; 8Department of Student Services, Cherokee County School District, 141 Twin Lake Road, Gaffney, SC 29341, USA; 9Department of Education, South Carolina Department of Health and Environmental Control, 2100 Bull St., Columbia, SC 29201, USA; 10Healthy Me—Healthy SC, Medical University of South Carolina, and Clemson University, 171 Ashley Avenue, Charleston, SC 29425, USA; 11Department of Pediatrics, College of Medicine, Medical University of South Carolina, 135 Rutledge Avenue, 3rd Floor, Charleston, SC 29425, USA

**Keywords:** human papillomavirus (HPV) vaccination, medically underserved, South Carolina

## Abstract

Human papillomavirus (HPV) infections are linked to at least six different types of cancer. The Medical University of South Carolina (MUSC) Hollings Cancer Center (HCC) and Department of Pediatrics leaders identified suboptimal rates of HPV vaccinations in rural and medically underserved communities in South Carolina (SC). To address this major public health problem in SC, they received funding from the HealthyMe/HealthySC (HMHSC) program and HCC to create a statewide community engagement-focused HPV Vaccination Van Program in October 2021. The Program provides HPV vaccinations and other childhood immunizations in school districts and HMHSC health clinics throughout SC, focusing on children aged 9–18 who are eligible for the U.S. Centers for Disease Control and Prevention’s Vaccines for Children Program. As of 14 December 2022, the Program administered vaccinations in 16 counties of SC to 552 participants, 243 of whom received HPV vaccinations and were predominantly female (57.2%), aged 4–18 (95.9%), and self-identified as White (44.0%), Black (33.2%), or Hispanic/Latino (15.1%). Most had Medicaid (53.1%)/no insurance coverage (25.1%). The Program is expected to expand as the Program’s relationship with SC’s school districts grows. The Program provides a model for delivering mobile HPV vaccinations to rural children, thus reducing their cancer risk.

## 1. Introduction

In 2019, an estimated 79 million people were infected with human papillomavirus (HPV) in the United States (U.S.), with an average of 14 million incident infections per year [1]. South Carolina (SC) has approximately 580 new HPV-attributable cancers each year [2]. Based on the currently available data, there are over 200 types of HPV viruses that have been identified, with 14 high-risk HPV types responsible for most HPV cancers [1]. It is well-established that HPV infection is directly linked with at least six types of cancers that include vulvar, penile, vaginal, anal, oropharyngeal, and, most noticeably, cervical cancer [3,4]. HPV is estimated to cause more than 90% of cervical and anal cancers; 70% of vaginal, vulvar, and oropharyngeal cancers; and 60% of penile cancers [3,4,5].

Each year in the U.S., approximately 47,199 HPV-associated cancers are diagnosed. Of these cancers, approximately 37,000 are estimated to be directly attributable to HPV infection (HPV-attributable cancers). A total of 58% of these HPV-attributable cancers affect females, with the remaining 42% affecting males [5]. With such a strong correlation between HPV and the various HPV cancers, it is critical that people get vaccinated against this dangerous infection. There are currently three licensed vaccines on the global market for HPV protection against the most common pathogenic types. The three main vaccines developed and approved to date are Gardasil^®^, Gardasil 9^®^, and Cervarix,^®^ (Merck & Co., Inc., Rahway, NJ, USA) each of which is delivered using intramuscular methods. In the U.S., only Gardasil 9^®^, which incorporates the coverage of additional HPV types compared with prior vaccines, is currently available and FDA-approved to be used for routine vaccines. Gardasil 9^®^ provides immunity for 9 different types of HPV, including 6, 11, 16, 18, 31, 33, 45, 52, and 58 [6]. The HPV vaccines prevent 70–90% of all HPV-related cancers [7,8].

The current recommendations from the CDC Advisory Committee on Immunization Practices indicate that the age range of HPV vaccinations is between 9 and 45 years old, with a target age of 11 to 12 years for optimal efficacy [9,10,11]. It is not clear if vaccination for individuals above 27 years of age would yield the same benefit as in younger populations, and is up to the clinician and the individual to determine the risk of potential future HPV exposure versus the benefits of initiation of the HPV vaccine in this age group [9].

The recommendation for the Gardasil 9^®^ vaccine includes two separate schedules for the age groups of 9 to 14 years and 15 to 45 years [9,10,11,12]. The specific timeframe for individual doses is described below.
For the two-dose series for an individual who initiates HPV vaccination between the ages of 9 and 14 years [10]:
a.Initiation of the first dose.b.The second dose should be given 6 months after the initiation of the first dose. (If administration of the second dose is less than 5 months after the first, a third dose should be given at least 4 months after the second dose.)
For the three-dose series for an individual who initiates HPV vaccination between the ages of 15 and 45 years [9,11]:
a.Gardasil 9^®^ should be administered at months 0, 2, and 6.

A common concern from the public is the efficacy and safety of HPV vaccines. In a previous study led by the CDC, Gardasil 9^®^ alone was studied in 15,000 females and males aged 9–26 for vaccine safety [13]. The result was a combination of seven Phase III clinical trials in which the vaccination was administered as a three-dose regimen, and all subjects received at least one dose or more. All of these studies reported that Gardasil 9^®^ was well-tolerated in the study participants. The most common adverse effects were an injection-site reaction, a headache, and fever [13].

The efficacy of Gardasil 9^®^ and other HPV vaccines has been consistently demonstrated across numerous studies, showing a high HPV vaccine efficacy for the prevention of HPV across the vaccine types covered, in situ carcinomas, and in situ adenocarcinomas. In addition, the vaccines have shown no signs of a waning immunity at least 10–12 years after vaccinations.

Despite these scientific data on the benefits of the HPV vaccine for HPV and cancer prevention as well as the excellent safety profile, the vaccination rates remain low. In a report from the Centers for Disease Control and Prevention (CDC) in 2021, the U.S. rate for first-dose HPV vaccine initiation was 76.9% among teenagers between the ages of 13 and 17 years [14]. In SC, the HPV vaccine initiation rate for the same age group was slightly higher at 81.8% (Figure 1) [14].

SC has more than 5 million residents, many of whom live in rural areas; 43% of SC’s 46 counties are classified as rural by the U.S. Census, and 15% of the state’s population lives in rural areas [15]. SC’s population demographics are typical of the Deep South region of the U.S. and primarily comprise White (68.6%) and Black (27.0%) residents, with a small but growing Hispanic/Latino population (6.0%) [16]. SC compares unfavorably with the U.S. averages in educational levels and per capita and median household incomes. Nearly 18% of the SC rural population has not completed high school, compared with 12% of the urban SC population [17]. The median household income for Black and Hispanic/Latino people is about USD 20,000 less than for White people, and the poverty rate in rural SC is 20.6% compared with 12.8% in urban areas of the state, both of which are higher than national averages. The poverty rate in SC is 150% higher for Black versus White residents [4,18]. Populations in rural areas have higher average death rates for all cancers combined compared with populations in urban counties. Finally, SC has significant racial and ethnic disparities in health insurance coverage. Nearly 11% of SC residents lack any form of health insurance [19] and only 22% of Black and 4% of Hispanic/Latino adults in SC have employer-covered health insurance compared with 69% of White adults. Additionally, 44% of Black and 41% of White residents in SC rely on Medicaid or public health insurance compared with 9% of Hispanic/Latino. Among SC’s uninsured population, 28% are White, 52% are Black, and 16% are Hispanic/Latino [20].

The lower rates of HPV vaccination in rural areas of SC prompted the Medical University of South Carolina (MUSC) Hollings Cancer Center (HCC), working closely with the MUSC Department of Pediatrics, to establish a statewide, community outreach and engagement-focused program called the HPV Vaccination Van Program. The Program was launched in October 2021 with funding from Healthy Me/Healthy SC (HMHSC), the MUSC HCC, and the MUSC Department of Pediatrics. HMHSC is a collaborative partnership between MUSC and Clemson University that aims to improve health care access and health inequities in rural and underserved communities of SC [21]. The mission of HMHSC aligns with the main goal of the HPV Vaccination Van Program, which is to promote and increase HPV vaccination rates in rural and medically underserved areas of SC. We aimed to evaluate the current data to assess the program’s effectiveness in meeting its goal of providing mobile HPV vaccinations to this population.

## 2. Methods

The HPV Vaccination Van Program staff include a program director, a medical director, a program manager, a scheduler/health educator, and a pediatric nurse. The Program team works closely with community leaders in rural and medically underserved regions of SC. These organizational leaders include school district leaders (superintendents, principals, nurses, teachers, etc.), ministerial leaders, and leaders of other community groups.

The Program staff also work with the Director of the SC Department of Health and Environmental Control (DHEC) Childhood Immunization Program, who linked the Program with the nurses at each of the schools in the participating counties of SC. SC DHEC’s interest stemmed from the fact that SC had lower vaccination coverage estimates than the U.S. rates for all CDC Advisory Committee on Immunization Practices-recommended adolescent vaccines [22], based on NIS children [23] and NIS teen national survey data [14], as shown in Figure 2.

Regular meetings are conducted between the HPV Vaccination Van Program staff and the community leaders to make introductions and to plan the HPV vaccination event in the leaders’ community settings. During each meeting, the Program staff describe the goals and objectives of the Program whilst also identifying and aligning with the goals of the community and its leaders.

One of the first HPV Vaccination Van Program events with a school district took place in Cherokee County, SC. The school district leaders in the Cherokee County School District identified the need for a town hall meeting prior to the HPV vaccine clinic in their district. The purpose of the town hall meeting was to help educate and debunk the myths about the HPV vaccine. The Cherokee County school district leaders worked with the HPV Vaccination Van Program team to develop the agenda and flier for the one-hour virtual town hall meeting, which took place on 21 February 2022 from 6–7 p.m.

The virtual town hall meeting agenda included a discussion by a school district science teacher who had been diagnosed with a high-risk HPV infection. She shared the invasive procedures she had been required to undergo as a result. The next speaker was her 14-year-old son, who described his decision-making process for receiving the HPV vaccine. The next two speakers were MUSC physicians, who described what HPV is and how the HPV vaccine works. This was followed by 30 min of questions and answers from the virtual audience. The metrics showed that, over the two days after the virtual town hall event, 811 people viewed the recorded video.

The following week, the HPV Vaccination Van Program team administered 57 HPV vaccinations to children in 3 of the schools in the Cherokee County School District. As the school district administrators accurately predicted, giving parents an opportunity to learn more about the HPV vaccine and to hear it described as a cancer prevention method for at least six different types of HPV-related cancers proved to be an effective method for engaging them.

The HPV Vaccination Van Program has offered the video to several other school districts across the state, who have posted it on their websites prior to the vaccination clinics at their schools. The HPV Vaccination Van Program staff members also participate in health fairs in different regions of SC. During each health fair, if participants are interested in receiving the HPV vaccine, the Program staff assess their eligibility. If the participants are eligible, the program nurse then administers the HPV vaccine (or other designated vaccine). With the help of SC DHEC, the HPV Vaccination Van Program has also connected with school nurses to schedule vaccine clinics in rural communities of SC.

### Analytical Methods

The sociodemographic characteristics of the participants were evaluated overall and by vaccination status (e.g., participants who received routine childhood vaccinations with and without the HPV vaccine). The descriptive statistics (number of cases; percentages) were used to describe the study results. The categorical variables were compared by chi-squared tests; Fisher’s exact tests were used when the cell counts were less than five. All analyses were performed using International Business Machines’ Statistical Package for the Social Sciences (SPSS), version 28.0.1 (SPSS, Inc., Chicago, IL, USA).

## 3. Results

Since the HPV Vaccination Van Program was launched in October 2021, the Program staff have traveled to the following 16 SC counties, as shown on the map in Figure 3: Anderson, Barnwell, Beaufort, Berkeley, Charleston, Cherokee, Colleton, Darlington, Dorchester, Georgetown, Horry, Jasper, Lancaster, Laurens, Saluda, and Williamsburg.

At each location, the HPV Vaccination Van Program has administered the HPV, Tdap, Meningitis ACWY, MMR-Varicella, DTaP-IPV, and Hepatitis A vaccines (see Table 1). The HPV Vaccination Van Program provides vaccinations for people between the ages of 9 and 45 years, focusing on children aged 9 to 18 years who are eligible for the CDC’s Vaccines for Children (VFC) Program because the HPV Vaccination Van Program’s medical director recommended that the Program provide all vaccines for which participating adolescents might be due. The VFC program also required that the Program include all adolescent vaccines. The vaccine-bundling approach is an established best practice for vaccine programs to offer individuals all recommended vaccines “in the same way, on the same day” to optimize the efficiency and benefits of vaccinations. The Program leadership emphasized to community-based physicians serving pediatric patients that the goal of the Program was to assist physicians with additional access to vaccine coverage for their patients rather than to replace the physicians as health care providers.

To achieve the goal of improving HPV vaccination rates, the Program team travels to elementary schools, middle schools, high schools, health fairs, and community events to administer the vaccine. As of 14 December 2022, the program team has delivered vaccines to a total of 552 participants (see Table 2). Of the 552 participants, 243 have received the HPV vaccine. The following results apply to these 243 participants who received the HPV vaccination. Approximately 96% of HPV vaccinations were given to participants between the ages of 4 and 18 years, 0.4% between the ages of 19 and 24 years, 0.4% between the ages of 25 and 34 years, 2.9% between the ages of 35 and 44 years, 0.4% between the ages of 45 and 64, and none between the ages of 65 and 81 years. In terms of gender, 139 females (57.2%) and 104 males (42.8%) received the HPV vaccine.

Of note, among the 243 participants who received the HPV vaccine, Table 2 shows that the majority (102; 44.0%) were White; 77 (33.2%) were Black. Furthermore, in terms of insurance status, among the 243 participants who received the HPV vaccine, the majority (114; 51%) had Medicaid; 61 (27.5%) had no insurance coverage. The 243 participants were from 26 counties in SC, although an accurate zip code was not available for 2 participants (Table 3).

## 4. Limitations

COVID-19 posed a significant limitation to the HPV Vaccination Van Program. Elementary, secondary, and high school administrators as well as school nurses were contacted by the Program staff about partnering with the Program. Initially, several school leaders expressed excitement about the Program, as many of their students were behind in their required vaccines due to delayed doctor visits as a result of COVID-19-related shutdowns. However, the nursing staff within the schools quickly became overwhelmed as the schools began to revert to in-person instruction. The school systems were initially more focused on the COVID-19 precautions associated with the re-opening the schools following the height of the pandemic, rather than on administering childhood immunizations. School staff also faced many challenges due to a lack of adherence to mask-wearing (which was not mandated in SC after 11 May 2021) among community members. A resurgence of COVID-19 cases among students and staff subsequently occurred. This caused severe understaffing, leading to several school closures; this delayed the start of the Program.

## 5. Discussion

The purpose of this study was to evaluate the first year of data from the MUSC HCC HPV Vaccination Van Program to assess the Program’s effectiveness in meeting its goal of increasing the number of people in SC who have received the HPV vaccine. MUSC HCC and MUSC Department of Pediatrics leadership developed the statewide, community outreach, and engagement-focused Program with funding from HMHSC and HCC to improve HPV and other childhood immunization rates in SC and to deliver evidence-based vaccine education. Along with providing the HPV vaccine, six other routine school-required vaccines, including Tdap, Meningitis ACWY, and Hepatitis A, are provided for those who are eligible to support a vaccine-bundling approach. Vaccinations are provided to people aged 9–45 years, focusing on children aged 9–18 years who are eligible for CDC’s VFC Program.

To date, the Program has been offered in 16 counties of SC. The HPV Vaccination Van Program strategically chose elementary, middle, and high school campuses as vaccination locations to increase the vaccination rates among children between the ages of 9 and 18 years old. Research has shown that this is the optimal age group to receive the HPV vaccine [6]. With important collaborations now established and expanding to support vaccination, it is anticipated that the number of vaccines can be dramatically increased over time.

Most of the participants served by the Program are insured by Medicaid, highlighting the fact that the Program has improved the vaccination rates among populations with lower financial resources in SC. The program participants who received vaccines from the HPV van were representative of SC’s demographics, and the results of the ongoing Program will be used to inform future HPV vaccination interventions in SC. The Program team aims to reach every county in SC.

The inaugural outing of the mobile HPV Vaccination Van Program took place in October 2021 at a health fair sponsored by HMHSC. Once the number of COVID-19 cases waned, school nurses were able to shift their focus back to getting the students up-to-date with their routine childhood vaccinations. In March 2022, the mobile HPV Vaccination Van Program began providing services to school districts throughout SC. To date, the Program has provided vaccination services to schools in 16 counties at several community sites.

## 6. Conclusions

The HPV infection is linked to at least 6 different types of cancer, and it causes over 580 new HPV cancers in SC each year [2]. The HPV Vaccination Van Program has successfully administered the HPV vaccine and other childhood vaccines to children in 16 counties of SC, thus reducing their cancer risk. The majority of the program participants were Medicaid recipients, which demonstrated the goal of the program in reaching medically underserved populations in SC. Despite facing challenges caused by implementing the program during the COVID-19 pandemic, the Program is proving to be successful in establishing partnerships with school districts and other community organizations in the state; the Program staff now receive calls inviting the HPV Vaccination Van Program team to community locales such as health care clinics serving rural and medically underserved people. Therefore, the Program staff anticipate a growth in the number of HPV vaccines that are delivered over the coming years with these established partnerships now in place. The Program recently initiated a partnership for promoting vaccines and providing vaccinations with PASOS in SC, an organization that employs a Community Health Worker/Promotore model statewide to serve Hispanic/Latino individuals and families with the goal of improving health education and awareness, advocacy, and resource navigation [24]. Future directions for this program include exploring how to better engage community providers with vaccinations through strategies such as academic detailing as well as providing further vaccine outreach and education among rural SC populations. The Program serves as a model for mobile HPV vaccination programs for rural and medically underserved children, thus reducing their cancer risk.

## Figures and Tables

**Figure 1 jcm-12-01362-f001:**
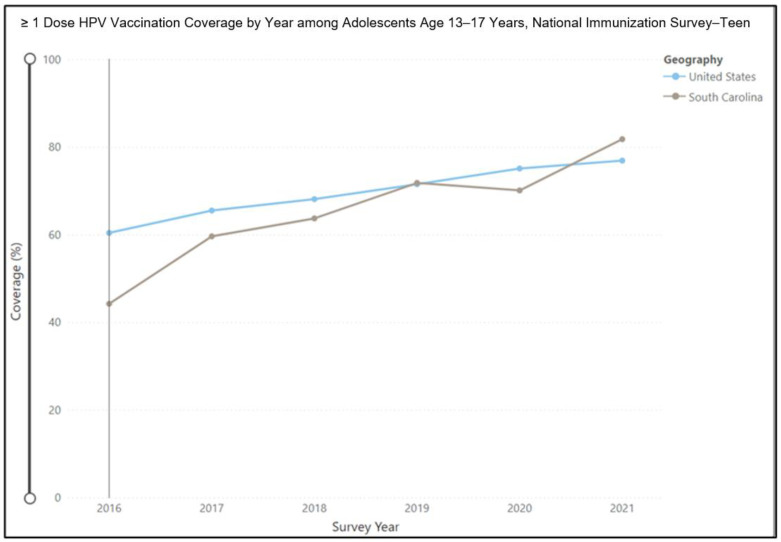
HPV vaccination rates in the United States and in South Carolina.

**Figure 2 jcm-12-01362-f002:**
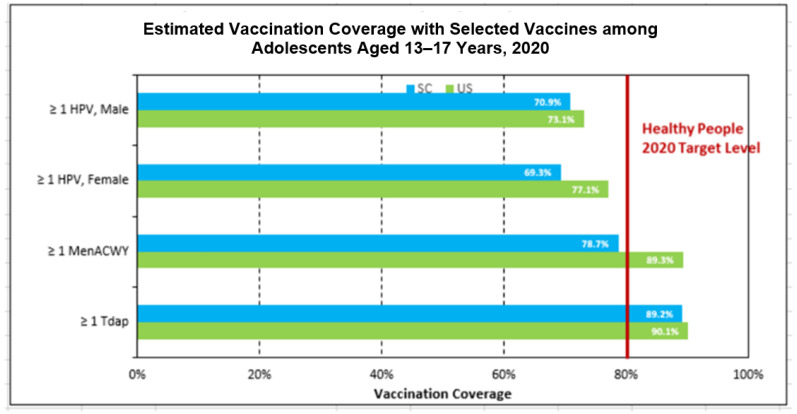
Estimated teen vaccination coverage in South Carolina and the United States, 2020.

**Figure 3 jcm-12-01362-f003:**
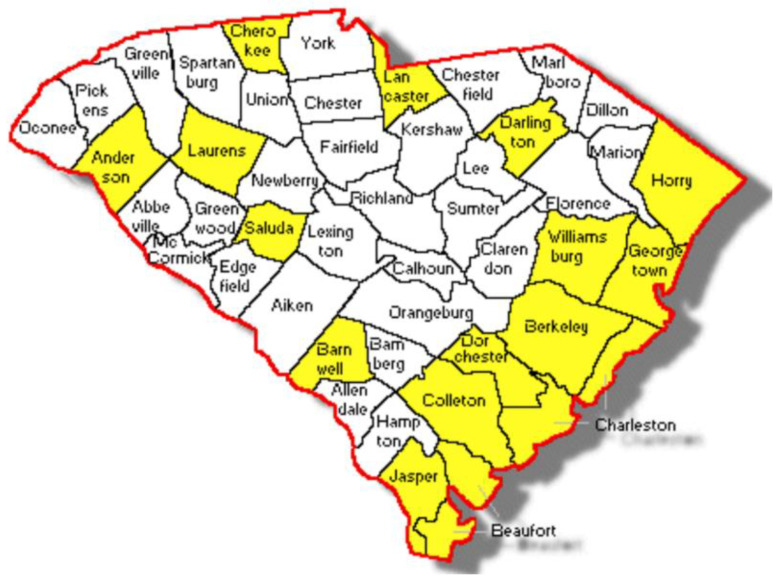
South Carolina counties served by the HPV Vaccination Van Program.

**Table 1 jcm-12-01362-t001:** Vaccinations and their abbreviations.

Vaccination	Vaccine Abbreviation
Human Papillomavirus	HPV
Tetanus, Diphtheria, Pertussis	Tdap
Combined Diphtheria, Tetanus, Pertussis, and Inactivated Poliomyelitis Vaccine	DTap-IPV
Meningococcal	Meningitis
Measles, Mumps, Rubella	MMR
Varicella	Chicken Pox
Hepatitis A	Hep A

**Table 2 jcm-12-01362-t002:** Sociodemographic characteristics of the HPV Van Vaccination Program participants overall and by receipt of routine childhood vaccinations with or without the HPV vaccine ^a^.

	Participants Receiving Routine Childhood Vaccinations (*n* = 309) *n* (%)	Participants Receiving Routine Childhood Vaccinations Including the HPV Vaccine (*n* = 243) *n* (%)	Total (*n* = 552) *n* (%)	*p*-Value
Gender				0.894
Female	175 (56.6)	139 (57.2)	314 (56.9)	
Male	134 (43.4)	104 (42.8)	238 (43.1)	
Age group (years)				<0.001
4–18	268 (86.7)	233 (95.9)	501 (90.8)	
19–24	0 (0)	1 (0.4)	1 (0.2)	
25–34	4 (1.3)	1 (0.4)	5 (0.9)	
35–44	4 (1.3)	7 (2.9)	11 (2.0)	
45–64	18 (5.8)	1 (0.4)	19 (3.4)	
64–81	15 (4.9)	0 (0)	15 (2.7)	
Race/ethnicity ^a^				0.59
American Indian	1 (0.3)	0 (0)	1 (0.2)	
Asian	5 (1.7)	3 (1.3)	8 (1.5)	
Black	137 (45.5)	77 (33.2)	214 (40.2)	
Hispanic/Latino	40 (13.3)	35 (15.1)	75 (14.1)	
Multiracial	11 (3.7)	7 (3.0)	18 (3.4)	
White	105 (34.9)	102 (44.0)	207 (38.8)	
Other race/ethnicity	1 (0.3)	2 (0.9)	3 (0.6)	
Other (Arab)	0 (0)	2 (0.9)	2 (0.4)	
Other (Middle Eastern)	0 (0)	1 (0.4)	1 (0.2)	
Egyptian	0 (0)	1 (0.4)	1 (0.2)	
Not listed	1 (0.3)	1 (0.4)	2 (0.4)	
Insurance status				0.017
Private	58 (18.8)	53 (21.8)	111 (20.1)	
Medicaid and Private	1 (0.3)	0 (0)	1 (0.2)	
Medicaid	153 (49.5)	129 (53.1)	282 (51.1)	
Medicare	10 (3.2)	0 (0)	10 (1.8)	
None	87 (28.2)	61 (25.1)	148 (26.8)	

^a^ Of the 552 participants, race/ethnicity was not listed for 2 participants (1 who received routine childhood vaccinations and 1 who received childhood vaccinations including the HPV vaccine); and was missing for 19 participants (8 who received routine childhood vaccinations and 11 who received childhood vaccinations including the HPV vaccine).

**Table 3 jcm-12-01362-t003:** County of residence of the HPV Van Vaccination Program participants overall and by receipt of routine childhood vaccinations with or without the HPV vaccine ^a^.

SC County of Residence of Each Program Participant	Participants Receiving Routine Childhood Vaccinations (*n* = 307), *n* (%)	Participants Receiving Routine Childhood Vaccinations Including the HPV Vaccine (*n* = 241), *n* (%)	Total ^a^ (*n* = 548), *n* (%)
Anderson	27 (8.8)	0 (0)	27 (4.9)
Bamberg	0 (0)	2 (0.8)	2 (0.4)
Barnwell	0 (0)	5 (2.1)	5 (0.9)
Beaufort	30 (9.8)	30 (12.4)	60 (10.9)
Berkeley	29 (9.4)	1 (0.4)	30 (5.5)
Charleston	5 (1.6)	5 (2.1)	10 (1.8)
Cherokee	18 (5.9)	73 (30.1)	91 (16.6)
Colleton	17 (5.5)	13 (5.4)	30 (5.5)
Darlington	22 (7.2)	16 (6.6)	38 (6.9)
Dorchester	43 (14.0)	9 (3.7)	52 (9.5)
Florence	2 (0.7)	2 (0.8)	4 (0.7)
Georgetown	43 (14.0)	0 (0)	43 (7.8)
Greenville	1 (0.3)	0 (0)	1 (0.2)
Hampton	2 (0.7)	4 (1.7)	6 (1.1)
Horry	1 (0.3)	1 (0.4)	2 (0.4)
Jasper	4 (1.3)	2 (0.8)	6 (1.1)
Lancaster	37 (12.1)	34 (14.1)	71 (13.0)
Laurens	4 (1.3)	2 (0.8)	6 (1.1)
Lexington	0 (0)	1 (0.4)	1 (0.2)
Newberry	0 (0)	2 (0.8)	2 (0.4)
Orangeburg	1 (0.3)	2 (0.8)	3 (0.5)
Richland	0 (0)	2 (0.8)	2 (0.4)
Saluda	7 (2.3)	9 (3.7)	16 (2.9)
Spartanburg	0 (0)	6 (2.5)	6 (1.1)
Sumter	9 (2.9)	20 (8.3)	29 (5.3)
Williamsburg	5 (1.6)	0 (0)	5 (0.9)

Abbreviations: HPV, human papillomavirus; SC, South Carolina. ^a^ There was a total of 552 program participants. However, the data for 4 participants were excluded from the Table 3 analyses because 1 participant who received routine childhood vaccinations resided in Escambia County, Florida (outside of South Carolina) and the zip codes were erroneous for 3 participants (1 received routine childhood vaccinations and 2 received childhood vaccinations including the HPV vaccine).

## Data Availability

The aggregated, de-identified data will be shared publicly through this and additional scientific publications. Thus, these data will be made publicly available and cited in accordance with journal guidelines.

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
