# Peer review of "Evaluation of the First-Year Data from an HPV Vaccination Van Program in South Carolina, U.S."

_jcm, 2023, doi:10.3390/jcm12041362_

Round 1

Reviewer 1 Report

-    Comments for the manuscript

1.      Implement strongly bibliography with peer review manuscript instead of multiple links to web sites

2.      ref 1 in not presented as the first ref, check and organize

3.      fig. leg of fig 1 must be correct, since it reports “HPV vaccination as causative agent of cancers…”

4.      Detail manufactured details of HPV vaccines, and the route of injection

5.      Include details of clinical trials cited

6.      Detail “national” al line 105, and the year of reference

7.      Include a closed brackets at line 177

8.      Remove ref 16

9.      “a” was not refers in Tab 4 and 5, except as foot note. Please revised tables

10.   Lines 217-224 reports data presented in tab 4. Lines can be revised ion order to avoid duplicate

11.   Discussion can be concluded highlighting the benefit of the study

Author Response

Comments and Suggestions for Authors

 Comments for the manuscript

Point 1.   Implement strongly bibliography with peer review manuscript instead of multiple links to web sites

Response 1. Thank you. We agree that adding original peer-reviewed scientific references will strengthen this manuscript. Thus, multiple websites that were previously listed as references have now been removed and replaced with scientific peer-reviewed published references. The revised references are noted throughout the document using the Track Changes format.

Point 2.   ref 1 in not presented as the first ref, check and organize

Response 2. Thank you. We agree, and we have corrected this error.  

Point 3. fig. leg of fig 1 must be correct, since it reports “HPV vaccination as causative agent of cancers…”

Response 3. Thank you. Figure 1 has been removed.

Point 4. Detail manufactured details of HPV vaccines, and the route of injection

Response 4. We concur with the reviewer. All injections are delivered using intramuscular methods (IM). Gardasil made by Merck, and Cervarix made by GlaxoSmithKline. The text of the manuscript was revised to denote the manufacturers of the HPV vaccines and the route of injection.

Point 5. Include details of clinical trials cited

Response 5. Thank you. Details have been added to describe additional detail of clinical trials that have evaluated HPV vaccination. Specifically, detail was added about the age group of participants and HPV vaccination delivery protocol.   

Point 6. Detail “national” al line 105, and the year of reference

Response 6. Throughout the manuscript, the word “national” was removed and replaced with “U.S.”.

Point 7. Include a closed brackets at line 177

Response 7. Thank you for picking up this typo error. This change was made.

Point 8. Remove ref 16

Response 8. Thank you. This reference was removed.

Point 9. “a” was not refers in Tab 4 and 5, except as foot note. Please revised tables

Response 9. We have added the footnotes “a” to the titles of Tables 2 and 3 (which replaced the former Tables 4 and 5). We believe that the footnotes accurately describe the data that are presented in the tables.   

Point 10. Lines 217-224 reports data presented in tab 4. Lines can be revised in order to avoid duplicate

Response 10. The text was revised to state: “Of note, among the 222 participants who received the HPV vaccine, Table 2 shows that the majority, 99 (46.9%) were White and  62 (29.4%) were Black. Furthermore, in terms of insurance status, among the 222 participants who received the HPV vaccine, the majority, 114 (51.4%), had Medicaid and 61 (27.5%) had no insurance coverage.”

Point 11. Discussion can be concluded highlighting the benefit of the study

Response 11. The Discussion section was revised, and Limitations and Conclusions sections were added. The Conclusions section highlights the benefit of the study.

Reviewer 2 Report

general remarks:

- HPV vaccination as a causative agent of the multiple types of cancer (Fig. 1) - how the HPV Vx could be a causative agent if the purpose of Vx is to protect against it? it's a major error that constitutes the basis of misunderstanding of the entire topic

- table 1 is taken 1:1 from CDC website, https://www.cdc.gov/hpv/parents/cancer.html

- the Vx role in HPV-derived cancer is not addressed enough, but it should be as it's the core thought described in the paper

- % of each cancer category should be calculated (table 1)

-HVP Vx efficacy and effectivness should be written in separate columns (table 2)

- no number of cases prevented calculated

- some figures/tables should be moved to supplementary materials (Fig. 4, fig. 5, table 5)

- language level should be more scientific

- the references list should be much broader, using much more scientific papers as the references than just links

- the references list should be unified to the same format

detailed remarks (refering to specific entries):

- 71 "strands" should be replaced with "types"

- 72 "vaccines" should be replaced with vaccination

- 77 "will" should be replaced with "would"

- 80 abbraviation not needed

- 84 point a is not needed

- 86-87 should be written in in parenthesis

- 89 point a is not needed

- 98 Number of HPV doses received; it's not a dosage, as taken from SmPC

- 100-102 please expatiate on this topic

- 111, 114 - these abbreviation should be given in the abstract

- 163 I'd rather consider preventing HVP Vx from at least 6 types of cancer than exactly 6 (the same referes to line 54)

- 165 "the parents" should be replaced with "them"

- 177-178 parenthesis is missing

- 193 - personal data not needed

- 211 - no serogropus for Meningococcus used; table 3 seems very basic for a peer-reviewed scientific journal

- 229 "rather than 520"

- 264-265 mask wearing is a part of C19 guidelines. Please change wording

Author Response

Submission Date

14 January 2023

Date of this review

16 Jan 2023 13:57:58

Reviewer 2

Open Review

English language and style

( ) English very difficult to understand/incomprehensible
( ) Extensive editing of English language and style required
(x) Moderate English changes required
( ) English language and style are fine/minor spell check required
( ) I don't feel qualified to judge about the English language and style

Yes

Can be improved

Must be improved

Not applicable

Does the introduction provide sufficient background and include all relevant references?

( )

( )

(x)

( )

Are all the cited references relevant to the research?

( )

(x)

( )

( )

Is the research design appropriate?

( )

(x)

( )

( )

Are the methods adequately described?

( )

( )

(x)

( )

Are the results clearly presented?

( )

(x)

( )

( )

Are the conclusions supported by the results?

( )

( )

(x)

( )

Comments and Suggestions for Authors

general remarks:

Point 1. HPV vaccination as a causative agent of the multiple types of cancer (Fig. 1) - how the HPV Vx could be a causative agent if the purpose of Vx is to protect against it? it's a major error that constitutes the basis of misunderstanding of the entire topic

Response 1. Thank you. Figure 1 has been removed.

Point 2. Table 1 is taken 1:1 from CDC website, https://www.cdc.gov/hpv/parents/cancer.html

Response 2. Table 1 was deleted. The following text was added:

The HPV vaccines prevent 70-90% of all HPV-related cancers.(Wang et al. 2020, PMID: 31812696; Harper and DeMars 2017, PMID 28442134).

Point 3. % of each cancer category should be calculated (table 1)

Response 3. Thank you. Table 1 was deleted.

Point 4. HVP Vx efficacy and effectiveness should be written in separate columns (table 2).

Response 4. Table 2 was deleted.

Point 5. no number of cases prevented calculated

Response 5. We have now added the statement listed in Response #2 above.

Point 6. some figures/tables should be moved to supplementary materials (Fig. 4, fig. 5, table 5)

Response 6. The original Table 1, Table 2, and the figure of the Program flyer were deleted.

Point 7. language level should be more scientific

Response 7. The language of the paper was revised to show a more scientific focus.

Point 8. the references list should be much broader, using much more scientific papers as the references than just links

Response 8. The references list was substantially revised to reflect the inclusion and citation of more scientific papers.  

Point 9. the references list should be unified to the same format

Response 9. Thank you. The references list has now been converted to a unified format

 Detailed Remarks:

Point 10. 71 "strands" should be replaced with "types"

Response 10. This change was made.

Point 11. 72 "vaccines" should be replaced with vaccination

Response 11. This change was made.

Point 12. 77 "will" should be replaced with "would"

Response 12. This change was made.

Point 13. 80 abbreviation not needed

Response 13. The abbreviation was deleted.

Point 14. 84 point a is not needed

Response 14. The points were changed to a bulleted list.

Point 15. 86-87 should be written in in parenthesis

Response 15. This change was made.

Point 16. 89 point a is not needed

Response 16. The points were changed to a bulleted list.

Point 17. 98 Number of HPV doses received: it's not a dosage, as taken from SmPC

Response 17. The original Table 2, which included the term “dosage” was deleted.

Point 18. 100-102 please expatiate on this topic

Response 18. This text was deleted.

Point 19. 111, 114 - these abbreviations should be given in the abstract

Response 19. The abstract was revised to include the definitions of the abbreviations.

Point 20. 163 I'd rather consider preventing HVP Vx from at least 6 types of cancer than exactly 6 (the same refers to line 54)

Response 20. The words “at least 6” were added to the text.

Point 21. 165 "the parents" should be replaced with "them"

Response 21. This change was made.

Point 22. 177-178 parenthesis is missing

Response 22. The missing right parenthesis was added.

Point 23. 193 - personal data not needed

Response 23. The personal data was removed.

Point 24. 211 - no serogroups for Meningococcus used; table 3 seems very basic for a peer-reviewed scientific journal

Response 24. We would prefer to keep Table 3 to help ensure that the paper is accessible to the audience of the Journal of Clinical Medicine, including researchers, scientists, and common readers. If required, we would be glad to include Table 3 as a supplemental table.

Point 25. Should be Men ACWY

Response 25. The text was changed to “Meningitis ACWY.”

Point 26. 229 "rather than 520"

Response 26. Thank you. The text in footnote “a” of the new Table 3 of the revised manuscript was revised to state: a There were a total of 552 program participants. However, the data for 4 participants were excluded from the Table 3 analyses because one participant who received routine childhood vaccinations resided in Escambia County, Florida (outside of South Carolina), and the zip codes were erroneous for 3 participants (1 received routine childhood vaccinations, and 2 received childhood vaccinations including the HPV vaccine).

Point 27. 264-265 mask wearing is a part of C19 guidelines. Please change wording.

Response 27. In South Carolina, U.S., a state mandate was issued in May of 2021 banning mask mandates for state employees, including state-funded school systems. If school employees and students choose to wear masks in the school setting, they are allowed to do so. However, many choose not to do so, resulting in repeated outbreaks of COVID-19 infections in the school systems.

Submission Date

14 January 2023

Date of this review21 Jan 2023 18:51:14

Round 2

Reviewer 2 Report

Having reviewed this paper I am still concerned of it's importance from the scientific point of view. For me it provides just a little, if any, relevant data. The resubmitted version includes many major changes, in line with my given instructions.

Author Response

Open Review

English language and style

( ) English very difficult to understand/incomprehensible
( ) Extensive editing of English language and style required
( ) Moderate English changes required
(x) English language and style are fine/minor spell check required
( ) I don't feel qualified to judge about the English language and style

Yes

Can be improved

Must be improved

Not applicable

Does the introduction provide sufficient background and include all relevant references?

( )

(x)

( )

( )

Are all the cited references relevant to the research?

( )

(x)

( )

( )

Is the research design appropriate?

( )

(x)

( )

( )

Are the methods adequately described?

( )

(x)

( )

( )

Are the results clearly presented?

( )

(x)

( )

( )

Are the conclusions supported by the results?

( )

(x)

( )

( )

Comments and Suggestions for Authors

  1. Having reviewed this paper I am still concerned of it's importance from the scientific point of view. For me it provides just a little, if any, relevant data.

Thank you for your second review of the paper. In response to your comment, to emphasize the scientific importance of the paper, additional text was added:

  • The abstract was revised to show the relationship between the HPV vaccine and cancer risk reduction: “Human papillomavirus (HPV) infections are linked to at least six different types of cancer.”
  • The abstract was also revised to include the following statement: “The Program provides a model for delivering mobile HPV vaccination to rural children, thus reducing their cancer risk.”
  • In the Introduction, the following text was added to show that South Carolina (U.S.) is a relatively poor and medically underserved state that is in great need of the HPV Vaccination Van Program: “SC has more than 5 million residents, many of whom live in rural areas; 43% of SC’s 46 counties are classified as rural by U.S. Census and 15% of the state population lives in rural areas.1 SC’s population demographics are typical of the Deep South region of the U.S. and are comprised primarily of White (68.6%) and Black (27.0%) residents, with a small but growing Hispanic/Latino population (6.0%).2 SC compares unfavorably with the U.S. averages in educational level and per capita and median household income. Nearly 18% of the SC rural population has not completed high school, compared with 12% of the urban SC population.3 The median household income for Black and Hispanics/Latino people is about $20,000 less than for White people, and the poverty rate in rural SC is 20.6%, compared with 12.8% in urban areas of the state, both of which are higher than national averages. The poverty rate in SC is 150% higher for Black versus White state residents.4 Populations in rural areas have higher average death rates for all cancers combined, compared to populations in urban counties. Finally, SC has significant racial and ethnic disparities in health insurance coverage. Nearly 11% of SC residents lack any form of health insurance6 and only 22% of Black and 4% of Hispanic/Latino adults in SC have employer-covered health insurance, compared to 69% of White adults. Additionally, 44% of Black people and 41% of White people rely on Medicaid or public health insurance, compared to 9% of Hispanics/Latino. Among SC’s uninsured population, 28% are White, 52% are Black, and 16% are Hispanic/Latino.7
  • The following text was added to the Conclusions section: “Therefore, the Program staff anticipate growth in the number of HPV vaccines that are delivered over the coming years, with these established partnerships now in place. The Program recently initiated a partnership for promoting vaccines and providing vaccinations with PASOS in SC, an organization that employs a Community Health Worker/Promotore model statewide to serve Hispanic/Latino individuals and families with the goal of improving health education and awareness, advocacy, and resource navigation 24. Future directions for this program will include exploring how to better engage community providers in vaccination through strategies such as academic detailing as well as providing more vaccine outreach and education among rural SC populations. The Program serves as a model for mobile HPV vaccination programs for rural and medically underserved children, thus reducing their cancer risk.”

  1. The resubmitted version includes many major changes, in line with my given instructions.

Thank you. We carefully responded to each of your previous comments.

Submission Date

14 January 2023

Date of this review

29 Jan 2023 18:50:41